# The Influence of Nutritional Status and Sleep Quality on Gustatory Function in Older Adults

**DOI:** 10.3390/medicina59010041

**Published:** 2022-12-26

**Authors:** Elif Esra Ozturk, Zeynel Abidin Ozturk

**Affiliations:** 1Department of Gastronomy and Culinary Arts, Faculty of Fine Arts and Architecture, Gaziantep Islam, Science and Technology University, Gaziantep 27010, Turkey; 2Department of Geriatrics, Medical Faculty, Gaziantep University, Gaziantep 27410, Turkey

**Keywords:** gustatory function, nutritional status, sleep quality, older adult

## Abstract

*Background and Objectives:* Age-related declines in taste function are common. Taste acuity can be affected by nutritional status and sleep quality. This research aims to examine the effect of nutritional status and sleep quality on gustatory function in community-dwelling older adults. *Materials and Methods*: This cross-sectional study included 119 community-dwelling older adults (50.4% of whom were female). The gustatory function was evaluated using four liquid taste solutions (sweet, bitter, sour, and salty) each at four different concentrations and the Mini Nutritional Assessment (MNA) and Pittsburgh Sleep Quality Index (PSQI) was applied. Additionally, anthropometric measurements were taken. *Results:* The mean scores on the gustatory test for the sweet, bitter, sour, and salty tastes were 2.11 ± 1.27, 2.12 ± 1.03, 2.28 ± 1.03, and 1.98 ± 1.41, respectively. There were significant differences according to gender, polypharmacy, nutritional status, and sleep quality in identifying sweet tastes (all *p* < 0.05). It was also found that females and participants without polypharmacy had better scores for bitter tastes. When the gustatory functions were evaluated according to BMI classification, it was determined that underweight participants had a higher sour taste score than the obese ones. Multiple regression analysis revealed that age, MNA score, PSQI score, and gender accounted for a total of 20.4% of the variance in the sweet taste score. *Conclusions:* Determining the relationship between taste function, nutritional status, and sleep quality in older adults is important in terms of developing new strategies for older adults who have these problems.

## 1. Introduction

The aging process is gradual, inherent, and universal; it occurs in all living forms as a direct result of the interaction between an individual’s genetics and their environment [1]. The dominant demographic phenomenon of the twenty-first century is population aging [2,3]. The United Nations (UN) estimates that there will be more than 1.5 billion people over 65 years of age in the world by 2050, which is more than double the current population [4]. This trend has led to the need for more studies on how to help older people maintain their health and reduce their risk of disease [5].

The gustatory (taste) and olfactory (smell) functions play crucial roles in nutritional health [6]. The senses detect environmental signals, encode and send them to the central nervous system, and provide information for perception [7]. Sensory functions, particularly gustatory and olfactory functions, influence appetite, choice, and intake [8,9,10,11].

Aging is characterized by a decline in overall sensory perception [12,13]. The most common causes of age-related decline in gustatory function are physiological changes, such as a decrease in the density of taste buds and papillae, a decrease in the dysfunction of taste receptor cells responsive to neural responses, difficulties in maintaining oral health and a decline in olfactory function, chronic disease and related polypharmacy [14,15]. Due to the decline in gustatory functions and the resulting shift to unhealthy eating habits, this could have severe health consequences, as it increases the risk of diet-related disorders, creating a vicious cycle of disease [14,16,17]. For example, a diminished sense of salty taste may lead individuals to season their food with excessive salt, thus increasing their risk of cardiovascular disease [17]. Furthermore, in older people, changes in taste perception can be a factor in decreased appetite and reduced food intake, which can lead to malnutrition [18,19]. It would appear that the sensory system not only contributes to certain disorders but is also affected by them. For example, individuals diagnosed with diabetes have been documented to experience taste disorders and impairments [20]. In people with diabetes, the number of gustatory anatomical structures that contain taste buds decreases, and the morphology and vascularization of these structures are altered [21].

Aside from changes in gustatory functions, aging is known to cause changes in sleep patterns (for example, extending the amount of time spent in bed without sleeping, having trouble falling asleep again, sleeping for less hours at night, having longer sleep latency, having slower waves and waking up earlier, and having more daytime sleepiness and fatigue) [22,23].

In light of the fact that older adults may exhibit changes in nutritional status, sleep quality, and gustatory system, the aim of the current study is to determine the influence of sleep quality and nutritional status on gustatory function in older adults.

## 2. Material and Methods

### 2.1. Study Design and Participants

In this cross-sectional study, 119 older adults were used from the outpatient geriatric clinic at Gaziantep University Şahinbey Research and Application Hospital. Individuals who agreed to participate in the study were required to fill out an informed consent form according to the Helsinki Declaration. Ethical approval of the study was obtained from the Ethics Board of Gaziantep Islam Science Technology University (Protocol no: 2022/119, date of approval 7 June 2022). Participants under 65 years of age, suffering from neurodegenerative diseases (e.g., Alzheimer’s disease, dementia, Parkinson’s disease), those with musculoskeletal conditions, physical disabilities, renal or liver diseases, the presence or previous history of cancer, mouth/facial pain, smokers, and those who were parenterally and/or enterally fed were excluded from the study. Data on general information, anthropometric measurements, Mini Nutritional Assessment, and Pittsburgh Sleep Quality Index were collected via face-to-face interviews by healthcare researchers. Moreover, a taste test was applied to each participant.

### 2.2. Anthropometric Measurements

A trained dietitian took anthropometric measurements (weight, height, waist, and mid-upper arm circumference (MUAC)) using standard measurements procedures [24]. Body weight was determined using an electronic scale to the nearest 0.1 kg. Stadiometer was used to measure height to the nearest 0.1 cm. An inflexible tape was used to measure waist circumference to the nearest 0.1 cm from middle point between lower rib bone and crystal iliac bone. In addition, the body mass index (BMI) was calculated by dividing weight (kg) by the square of height (m^2^) and classified according to the World Health Organization, Geneva, Switzerland [25].

### 2.3. Assessment of Nutritional Status

The Mini Nutritional Assessment (MNA) [26], which was developed especially for the older adults and consists of 18 questions in four sections: anthropometric measurements, global assessment, dietary assessment, and subjective assessment, was used for nutritional screening. The nutritional status was evaluated using the following cutoff values: less than 17 indicates malnutrition, 17–23.5 indicates risk for malnutrition, and 24–30 indicates normal nutritional status.

### 2.4. Sleep Quality

The Pittsburgh Sleep Quality Index (PSQI) is a self-reported questionnaire used to evaluate sleep quality. In this study, the validated Turkish version of the PSQI was used [27]. The PSQI consists of seven subscales (subjective sleep quality, sleep latency, sleep duration, habitual sleep activity, sleep disturbance, use of sleeping medications, and daytime dysfunction), and each subscale score ranges from 0 to 3. The total score on the PSQI is composed of the sum of the scores of the seven subscales, which range from 0 to 21. A total score of 5 or less indicates ‘good’ sleep quality, while a total PSQI score more than 5 indicates ‘poor’ sleep quality [27,28].

### 2.5. Gustatory Function

The gustatory function was assessed as previously described [29,30]. The method consists of four liquid solutions at four concentrations for taste: salty (NaCl 0.25, 0.1, 0.04, and 0.016 g/mL); sweet (sucrose 0.4, 0.2, 0.1, and 0.05 g/mL); sour (citric acid 0.075, 0.041, 0.0225, and 0.0125 g/mL); and bitter (quinine hydrochloride 0.0015, 0.0006, 0.0002, and 0.0001 g/mL).

Solutions were applied to the back of the tongue with a dropper (three drops) containing one drop of the tastant and two drops of distilled water. The participant selected one of four tastes: salty, sweet, sour, or bitter. The gustatory score was defined as the number of correctly identified tastes for each taste. The participant got 1 point for each correctly identified taste and 0 points for incorrect responses, which included not identifying the taste or confusing it with another. The scores for each taste ranged from 0 to 4.

The administration order was randomized through simple randomization trials, and the tastants were evaluated at increasing concentrations. To avoid interactions between gustatory stimuli, the individual took a sip of water between tests. All solutions were stored in amber glass bottles without visible labels, and only fresh tastants, made daily, were used.

### 2.6. Statistical Assessments

The data was evaluated using Statistical Package for Social Sciences (SPSS) version 23.0 (IBM Corp, Armonk, NY, USA). Graphs were created using GraphPad Prism 9 (GraphPad Software, San Diego, CA, USA). Visual and analytical methods were used to analyze the normality of the data. Data were reported as mean (X), and standard deviation (SD), or number and percentage for continuous and categorical variables, respectively. Differences between groups were assessed using the independent samples t-test for two groups or the one-way analysis of variance (ANOVA) for three groups. A forward multiple linear regression analysis was applied using gustatory score of each taste quality as the dependent variable and age, gender, BMI, number of medication intake, MNA score, PSQI score, as independent variables. All variables that were significant at the level of 0.10 were included in the model. A *p*-value < 0.05 was considered statistically significant.

## 3. Results

The general characteristics of the participants are shown in Table 1. The participants were 59 males (49.6%) and 60 females with a mean age of 74.57 ± 6.92 years. In the study, 26.9% of the participants had hypertension, and 19.8% had coronary artery disease. In addition, 15.9% of the participants did not have any disease. Of the participants, 39.4% (*n*:47) were taking antihypertensive medications, 23.5% (*n*:28) oral antidiabetic medication, and 3.4% (*n*:4) received insulin treatment. The proportion of those using anticoagulant and antiaggregant medications among participants was 8.4% (*n*:10) and 29.4% (*n*:35), respectively. More than half of the individuals (58.8%, *n*:70) were taking proton pump inhibitors, while 14.3% (*n*:17) were using anti-osteoporotic medications. Polypharmacy, which was defined as the use of five or more medications, was found to be prevalent among participants in this study at 47.9%. Most of the participants were pre-obese or obese (27.7% and 33.7%, respectively). The mean BMI of the participants was 27.76 ± 4.62 kg/m^2^. According to the MNA, 13.4% of the older adults had malnutrition, 47.9% were at risk of malnutrition, and 38.7% had a normal nutritional status. The mean PSQI scores were 5.15 ± 2.93. The mean scores on the gustatory test for the sweet, bitter, sour, and salty tastes were 2.11 ± 1.27, 2.12 ± 1.03, 2.28 ± 1.03, and 1.98 ± 1.41, respectively.

Figure 1 displays the number of correct identifications in the study according to gender. Female participants had higher taste scores than their men counterparts for sweet [2.38 ± 1.32 vs.1.83 ± 1.16], bitter [2.31 ± 1.06 vs. 1.93 ± 0.96], sour [2.46 ± 0.94 vs. 2.10 ± 1.07)], and salty [2.10 ± 1.54 vs. 1.86 ± 1.26] tastes (*t*_sweet_= −2.414; p_sweet_ = 0.017, *t_bitter_*= −2.064; p_bitter_ = 0.040, *t_sour_*= −1.963; p_sour_ = 0.046, *t_salty_*= −0.908; p_salty_ = 0.366).

Table 2 summarizes the gustatory functions of the groups. BMI classification significantly affected the sour taste score (*p* < 0.05). The post hoc comparison indicated that the underweight group had higher sour taste scores than the obese group. However, there were no significant differences between sweet, bitter, and salty taste scores based on BMI classification (*p* > 0.05). The sweet and bitter taste scores of the participants with polypharmacy were significantly lower in the participants without polypharmacy (*p* < 0.001, *p* < 0.05 respectively). The normal nutritional status group had a significantly higher sweet taste score than the malnutrition risk and malnutrition groups (*p* < 0.001) while there were no significant differences for salty, sweet, and bitter tastes between the groups. Those individuals who reported having poor sleep quality had a significantly lower sweet taste score than those participants who reported having good quality of sleep (*p* < 0.05). (Table 2).

Furthermore, when investigating the link between PSQI subscales and gustatory functions, the correlation between sleep duration and sweet and bitter taste scores was positive and significant (*r* = 0.285, *p* = 0.002; *r* = 0.243, *p* = 0.008, respectively). All taste scores were not significantly associated with subjective sleep quality, sleep latency, sleep duration, habitual sleep activity, sleep disturbance, use of sleeping medications, and daytime dysfunction (data not shown).

Age, MNA score, PSQI score, and gender together accounted for 20.3% of the variation in the sweet taste score in a multiple linear regression analysis. The females had a higher taste score for the sweet taste, which was also positively correlated with the MNA score and negatively correlated with age and PSQI score. Age was the only variable that significantly explained the bitter and salty taste scores (5.1% and 7.6%, respectively) in the model. Age and BMI were negatively related to the sour taste score and contributed to 11.4% of the model’s ability to predict the sour taste score (Table 3).

## 4. Discussion

In the current study, how nutritional status and sleep quality affected gustatory function were evaluated in 119 older adults, analyzing gender, age, BMI, and number of medications intake.

Taste is an extremely complicated modality that is influenced by a variety of internal and external circumstances. There are substantial differences between mental and physical impairments in older adults [31,32]. Age and gender were important predictors of performance in gustatory identification [3]. In line with earlier studies [3,33,34], females performed better in correctly identifying sweet, bitter, and sour tastes in this study. These are thought to occur at the level of taste buds. Sex steroid hormones may modulate taste processing in the brain since receptors for sex hormones appear to be prominent in several nuclei associated with central gustatory pathways. In particular, estrogen levels can affect taste-elicited activity in the periphery and the brainstem, especially in the limbic pathway [35]. This study demonstrated a negative association between BMI and the sour taste score but not sweet, salty, or bitter. In their study, Fernandez-Garcia et al. found that all taste strip scores were negatively correlated with BMI [36]. According to the findings of another study, a higher body mass index was associated with a lower taste sensitivity [37]. High taste acuity may help to reduce dietary overconsumption and the risk of obesity, whereas gustatory dysfunction may lead to a high-calorie diet and obesity [38]. Furthermore, the increase in the use of medication among older adults can also alter oral diseases that affect gustatory functions [39]. Certain medications, particularly when many medications are taken for treatment, induce taste changes in older adults [40]. Several medicines, including some antimycotics, antibiotics, anti-inflammatory agents, antihypertensive medications, immunosuppressants, antihyperlipidemic medications, endocrine medications, neurologic medications, and psychiatric drugs, have been shown to negatively affect taste perception and eating patterns [7,41,42]. Medications can influence the perception in taste of older people by impacting the neurological system, such as peripheral receptors or the brain. Medications can also cause dry mouth syndrome by modifying salivary flow rate and buffering [39,43]. The results of this study showed that older people who used five or more medications a day may have lower bitter and sweet taste identification scores.

Reduced taste and impaired gustatory sensitivity in older adults may contribute to their loss of appetite, which can lead to anorexia, weight loss and malnutrition [3,44]. On the other hand, poor nutritional status may cause a decrease in gustatory capacity [45]. There is evidence that, regardless of whether they are primary or secondary, nutritional deficiencies can alter taste perception or worsen the effects of aging [46]. Toffanello et al. found a link between malnutrition or even the possibility of malnutrition and decreased sensitivity to sour taste [15]. The findings of this research indicated that there was a positive correlation between the sweet taste score and the MNA score. However, further study is necessary in order to understand the precise nature of this association.

A limited number of studies have examined the effect of sleep quality or sleep insufficiency on gustatory function. In a study conducted by Gao et al., they reported that poor overall sleep quality was associated with an increased risk of impaired taste perception [47]. Another study found that the duration of sleep had a positive effect on the sensitivity to sweet taste in older adults [22]. In another study, taste sensitivity did not change after a single night of getting less than 7 h of sleep. This suggests that more than one night of changing sleep may be needed to change taste sensitivity [48]. In this study, poor sleep quality was associated with a decreased sweet taste score or the ability to correctly identify sweet taste among older participants. Furthermore, the sleep duration score was positively and significantly correlated with the sweet and bitter taste score. The results showed that longer sleep duration was significantly linked to a better ability to identify sweet and bitter tastes. Although the precise mechanisms are unclear, it is certain that sleep plays a variety of roles in biological homeostasis, especially in the endocrine system, which affects taste sensitivity, especially in older people [22,49,50,51,52,53,54]. Lack of sleep combined with exposure to nocturnal light can raise postprandial glucagon-like peptide-1 (GLP-1), which generally maintains or improves the ability to taste sweet and citric acid [50,53]. Leptin, on the other hand, can suppress certain sweet reactions by activating the K1 outward currents [51]. A short duration of sleep was associated with a reduction in leptin levels, which may explain why people seek sweet foods more frequently [54]. It is hypothesized that serotonin plays a role in the transduction of taste signals. In addition, it has been demonstrated that sleep disruption increases the stress response, which may influence extracellular serotonin levels. Therefore, a change in serotonin levels due to stress might be an additional explanation for the association between sleep disorders and altered taste perception [52].

Furthermore, it has been confirmed that as people age, difficulties in identifying all the qualities of the taste are experienced. Some researchers found that sweet taste sensitivity decreased with age [55,56,57,58]. Some others have reported a decrease in sour sensitivity with aging [55,59,60,61]. Similarly, many reports suggest a decline in salt sensitivity [55,59,61,62] and bitter sensitivity [55,59,60,61,62,63] with increasing age. In this study, the multivariate logistic regression analysis found that age was negatively associated with the scores for sweet, bitter, salty, and sour tastes; these findings are consistent with previous findings. In general, gustatory functions deteriorate in older people, which may be related to diseases of the central nervous system and the endocrine system [64,65]. According to a recent study conducted in the European population, aging is associated with a decrease in the perception of all tastes [66]. In their study, Martelli et al. found that aging impaired the identification of sweet and salty tastes [22]. Although taste loss is increasingly prevalent in the aged, the precise reasons for these abnormalities remain unknown.

There are certain limitations to this report. The study lacked a control group of younger participants and did not consider the effect of olfaction on gustatory qualities. It should also be highlighted that the cross-sectional methodology and the small sample size are limitations. The cross-sectional method of the study only gave a snapshot of the situation. Because of this, it is not possible to generalize the results of this study to all older individuals. Additionally, poor oral health, such as salivation disorders, can have a significant influence on gustatory function. As a result, oral circumstances must be examined before testing gustatory performance in research. Furthermore, we did not evaluate the umami flavor, which has been linked to nutritional status. Despite its limitations, the study’s findings are noteworthy since they are the first to assess taste identification performance, as well as the impact of nutritional status and sleep quality on gustatory function in the Turkish older population.

## 5. Conclusions

As a result, this study found that nutritional condition and sleep quality were related to the ability to identify sweet taste. Furthermore, the current study found that age was a significant predictor of all taste scores. Understanding the relationship between taste function, sleep quality, and nutritional condition is critical to a healthier and better aging process.

## Figures and Tables

**Figure 1 medicina-59-00041-f001:**
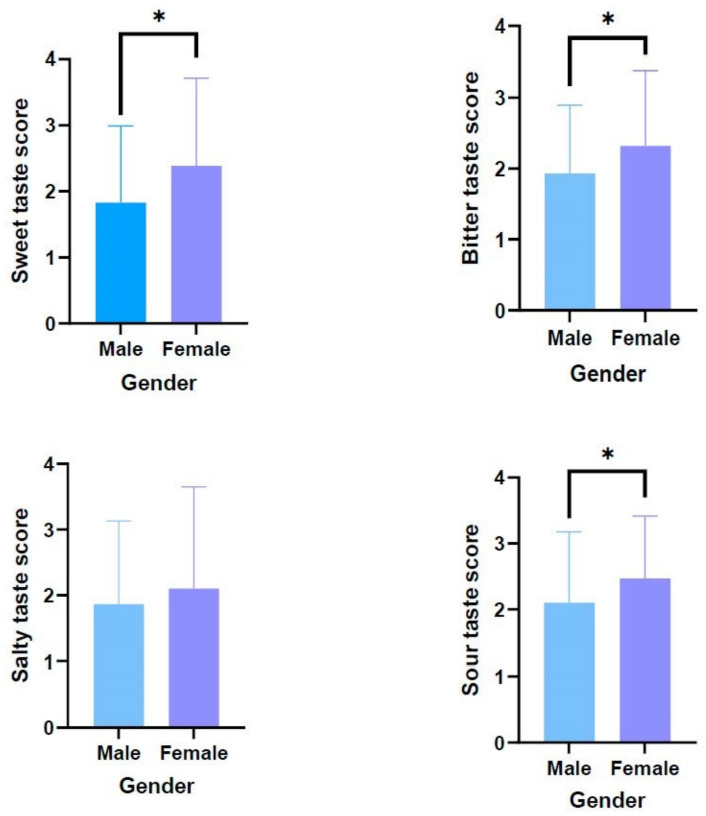
Differences in taste score by gender. The findings are presented as X ± SD. The use of a single asterisk (*) denotes the existence of statistically significant differences between the groups (*p* < 0.05).

**Table 1 medicina-59-00041-t001:** General characteristics of the participants.

	Total	
	*n*	%
Gender		
Male	59	49.6
Female	60	50.4
Age		
65–74	67	56.3
75–84	38	31.9
85^+^	14	11.8
Age (years) (X ± SD)	74.57 ± 6.92	
Diseases		
Hypertension	47	26.9
Diabetes Mellitus	32	18.2
Coronary Artery Disease	33	19.8
Gastrointestinal Diseases	13	7.4
Osteoporosis	22	12.6
None	28	15.9
Polypharmacy		
Yes	57	47.9
No	62	52.1
Number of medications (per day) (X ± SD)	3.84 ± 2.75	
BMI classification		
Underweight	17	14.3
Normal	29	24.4
Pre-obese	33	27.7
Obese	40	33.6
Weight (kg) (X ± SD)	74.70 ± 11.90	
BMI (kg/m^2^) (X ± SD)	26.75 ± 5.11	
Waist circumference (cm) (X ± SD)	102.46 ± 11.70	
MUAC (cm) (X ± SD)	29.58 ± 4.43	
Nutritional status		
Malnutrition	16	13.4
Risk of malnutrition	57	47.9
Normal	46	38.7
MNA Score (X ± SD)	21.63 ± 3.65	
Sleep Quality		
Good	65	54.6
Poor	54	45.4
PSQI total score (X ± SD)	5.15 ± 2.93	
Gustatory Function (X ± SD)		
Sweet taste score	2.11 ± 1.27	
Bitter taste score	2.12 ± 1.03	
Sour taste score	2.28 ± 1.03	
Salty taste score	1.98 ± 1.41	

BMI: body mass index, MUAC: mid-upper arm circumference MNA: Mini Nutritional Assessment, PSQI: Pittsburgh Sleep Quality Index.

**Table 2 medicina-59-00041-t002:** Comparison of gustatory functions in older participants in terms of BMI classification, polypharmacy, nutritional status, and sleep quality.

	Sweet Taste Core	Bitter Taste Score	Sour Taste Score	Salty Taste Score
Polypharmacy *				
Yes	1.50 ± 1.19	1.89 ± 1.03	2.12 ± 1.04	1.95 ± 1.13
No	2.66 ± 1.09	2.34 ± 0.96	2.44 ± 0.95	2.02 ± 1.61
Sig	−4.735	−2.356	−1.455	−0.768
*t*	<0.001	0.020	0.149	0.463
BMI **				
Underweight	2.59 ± 1.7	2.06 ± 1.25	2.94 ± 0.83 ^a^	2.12 ± 1.8
Normal	2.02 ± 1.36	2.07 ± 1.1	2.28 ± 1.1 ^ab^	2.41 ± 1.15
Pre-obese	2.24 ± 1.23	2.36 ± 0.93	2.45 ± 0.79 ^ab^	1.94 ± 1.37
Obese	1.88 ± 0.99	2.00 ± 0.96	1.88 ± 1.07 ^b^	1.65 ± 1.41
Sig	0.229	0.477	0.002	0.165
F	1.459	0.836	5.259	1.730
Nutritional status **				
Malnutrition	1.12 ± 1.50 ^a^	1.56 ± 0.96 ^a^	2.19 ± 1.03	1.87± 1.71
Risk of malnutrition	2.00 ± 1.25 ^b^	2.15 ± 1.11 ^ab^	2.31 ± 1.05	1.98 ± 1.43
Normal	2.58 ± 0.98 ^c^	2.28 ± 0.87 ^b^	2.39 ± 1.02	2.02 ± 1.30
Sig	<0.001	0.051	0.690	0.939
F	9.383	3.058	0.373	0.063
Sleep quality *				
Good (PSQI ≤5)	2.40 ± 1.25	2.13 ± 1.01	2.26 ± 1.04	1.93 ± 1.43
Poor (PSQI >5)	1.75 ± 1.21	2.12 ± 1.03	2.31 ± 1.00	2.05 ± 1.39
Sig	0.006	0.973	0.779	0.613
*t*	2.810	−0.034	−0.281	−0.510

* Independent sample *t* tests; ** one-way ANOVA, different lower letters in the same column indicate a statistical difference among the groups, according to post hoc test. Taste scores: sweet taste score ranges from 0 to 4, bitter taste score ranges from 0 to 4, sour taste score ranges from 0 to 4, salty taste score ranges from 0 to 4. The higher the score, the better gustatory function in each of taste quality.

**Table 3 medicina-59-00041-t003:** Determinants of the Taste Scores.

	β^1^ (95% CI)	SE	β^2^	*t*	*p*	R^2^
Sweet taste						
Model 1						0.089
Constant	6.39 (3.99–8.79)	1.211		5.276	<0.001	
Age	−0.06 (−0.09–−0.03)	0.016	−0.312	−3.549	0.001	
Model 2						0.130
Constant	4.15 (1.23–7.07)	1.475		2.812	0.006	
Age	−0.05 (−0.08–−0.02)	0.016	−0.271	−3.099	0.002	
MNA	0.08 (0.02–0.14)	0.030	0.223	2.547	0.012	
Model 3						0.168
Constant	4.30 (1.44–7.16)	1.443		2.981	0.004	
Age	−0.05 (−0.08–−0.02)	0.016	−0.253	−2.954	0.004	
MNA	0.08 (0.02–0.14)	0.030	0.233	2.723	0.007	
PSQI	−0.09 (−0.16–−0.02)	0.037	−0.212	−2.510	0.013	
Model 4						
Constant	3.15 (0.19–6.10)	1.490		2.111	0.037	0.203
Age	−0.04 (−0.07–0)	0.016	−0.195	−2.239	0.027	
MNA	0.09 (0.03–0.15)	0.029	0.254	3.019	0.003	
PSQI	−0.11 (−0.18–−0.03)	0.036	−0.242	−2.899	0.004	
Gender	0.53 (0.10–0.97)	0.219	0.211	2.444	0.016	
Bitter taste						0.051
Model 1						
Constant	4.82 (2.84–6.80)	0.999		4.831	<0.001	
Age	−0.03 (−0.06–−0.10)	0.013	−0.243	−2.713	0.008	
Sour taste						
Model 1						0.065
Constant	3.94 (2.85–5.03)	0.553		7.132	<0.001	
BMI	−0.06 (−0.10–−0.02)	0.020	−0.270	−3.036	0.003	
Model 2						0.0114
Constant	6.51 (4.36–8.65)	1.084		6.004	0.000	
BMI	−0.06 (−0.10–−0.02)	0.020	−0.262	−3.025	0.003	
Age	−0.04 (−0.06–−0.01)	0.013	−0.237	−2.728	0.007	
Salty taste						0.076
Model 1						
Constant	6.38 (3.70–9.07)	1.354		4.716		
Age	−0.06 (−0.09–−0.02)	0.018	−0.289	−3.265	0.001	

BMI: body mass index, MNA: Mini Nutritional Assessment, PSQI: Pittsburgh Sleep Quality Index, β^1^: unstandardized coefficient; β^2^: standardized coefficient.

## Data Availability

The data presented in this study are available on request from the corresponding author.

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
