# Peer review of "The Influence of Nutritional Status and Sleep Quality on Gustatory Function in Older Adults"

_medicina, 2022, doi:10.3390/medicina59010041_

Round 1

Reviewer 1 Report

Dear Authors

I REPORT THE CRUCIAL LIMITATIONS FOR YOUR PAPER:

1. IN THE PAPER ISN'T INDICATED THE PRESENCE OF CKD, IN PARTICULAR 4-5 CKD STAGE THAT CAN INFLUENCE THE RESULTS.

2 ISN'T REPORTED THE DRUGS TYPE, FOR EXAMPLE ANTIBIOTICS, ANTIDIABETIC AND ETC

3. ISN'T REPORTED THE WATER INTAKE AND HYDRATATION STATW

PLEASE DISCUTE THESE POINTS.

THANK YOU

Author Response

Dear Editor and Dear Reviewer

We greatly appreciate your helpful comments to improve our manuscript.  All the suggested revisions were completed on manuscript and replied here.

Yours Faithfully

Reviewer report:

I REPORT THE CRUCIAL LIMITATIONS FOR YOUR PAPER:

Point 1

IN THE PAPER ISN'T INDICATED THE PRESENCE OF CKD, IN PARTICULAR 4-5 CKD STAGE THAT CAN INFLUENCE THE RESULTS.

Response 1

The distribution of participating individuals according to their diseases is added to Table 1 as numbers and percentages. Diseases of the participants; Hypertension, Diabetes Mellitus, Coronary Artery Disease, Gastrointestinal Diseases, Osteoporosis.

In addition, those with renal or liver diseases or the presence or previous history of cancer were not included in the study and added to the exclusion criteria.

Diseases

n

%

Hypertension

47

26.9

Diabetes Mellitus

32

18.2

Coronary Artery Disease

33

19.8

Gastrointestinal Diseases

13

7.4

Osteoporosis

22

12.6

None

28

15.9

Point 2

ISN'T REPORTED THE DRUGS TYPE, FOR EXAMPLE ANTIBIOTICS, ANTIDIABETIC AND ETC

Response 2

The drug types used are added to the results section.

‘In the study, 26.9% of the participants had hypertension; 19.8% had coronary artery dis-ease. In addition, 15.9% of the participants did not have any disease. Of the participants 39.4% (n:47) were taking antihypertensive medications, 23.5% (n:28) oral antidiabetic medication and 3.4% (n:4) received insulin treatment. The proportion of those using anti-coagulant and antiaggregant medications among participants was 8.4% (n:10) and 29.4% (n:35), respectively. More than half of the individuals (58.8%, n: 70) were taking proton pump inhibitors, while 14.3% (n:17) were using anti-osteoporotic medications.’

Point 3

ISN'T REPORTED THE WATER INTAKE AND HYDRATATION STATW

PLEASE DISCUTE THESE POINTS.

Response 3

The nutritional status of the participants was evaluated according to the mini-nutritional evaluation. In this context, the daily fluid intake of individuals was evaluated. However, daily water consumption was not evaluated.

Reviewer 2 Report

This is a very interesting paper and would be helpful to both researchers and clinicians . However ,  it would be important to suggest an update of the literature on sense of taste and not only polypharmacy but the presence of chronic non-communicative diseases , including olfactory function which is not examined in this case. These limitations are described by the authors very well. 

Author Response

Dear Editor and Dear Reviewer

We greatly appreciate your helpful comments to improve our manuscript.  All the suggested revisions were completed on manuscript and replied here. Additions in the manuscript text were written in blue colour. 

Yours Faithfully,

Reviewer report:

Point: This is a very interesting paper and would be helpful to both researchers and clinicians.  However, it would be important to suggest an update of the literature on sense of taste and not only polypharmacy but the presence of chronic non-communicative diseases, including olfactory function which is not examined in this case. These limitations are described by the authors very well. 

Response: The introduction part of the study is restructured. Its relationship with diseases is stated in accordance with the current literature.

The updated version of the introduction is given below:

‘The aging process is gradual, inherent, and universal; it occurs in all living forms as a direct result of the interaction between an individual's genetics and their environment (1). The dominant demographic phenomenon of the twenty-first century is population aging (2, 3). The United Nations (UN) estimates that there will be more than 1.5 billion people over 65 years of age in the world by 2050, which is more than double the current population(4). This trend has led to the need for more studies on how to help elder people maintain their health and reduce their risk of disease (5).   

The gustatory (taste) and olfactory (smell) functions play crucial roles in nutritional health (6). The senses detect environmental signals, encode and send them to the central nervous system, and provide information for perception (7). Sensory functions, particularly gustatory and olfactory functions, influence appetite, choice, and intake (8-11).

Aging is characterized by a decline in overall sensory perception (12, 13). The most common causes of age-related decline in gustatory function are physiological changes such as a decrease in the density of taste buds and papillae, a decrease in the dysfunction of taste receptor cells responsive to neural responses, difficulties in maintaining oral health and a decline in olfactory function, chronic disease and the related polypharmacy (14, 15). Due to the decline in gustatory functions and the resulting shift to unhealthy eating habits, this could have severe health consequences, as it increases the risk of diet-related disorders, creating a vicious cycle of disease (14, 16, 17). For example, a diminished sense of salty taste may lead individuals to season their food with excessive salt, thus increasing their risk of cardiovascular disease (17). Furthermore, in elderly people, changes in taste perception can be a factor in decreased appetite and reduced food intake, which can lead to malnutrition (18, 19). It would appear that the sensory system not only contributes to certain disorders, but is also affected by them. For example, individuals diagnosed with diabetes have been documented to experience taste disorders and impairments (20). In people with diabetes, the number of gustatory anatomical structures that contain taste buds decreases, and the morphology and vascularization of these structures are altered (21).

Aside from changes in gustatory functions, ageing is known to cause changes in sleep patterns (for example, extending the amount of time spent in bed without sleeping, having trouble falling asleep again, sleeping for less hours at night, having longer sleep latency, having slower waves and waking up earlier, and having more daytime sleepiness and fatigue) (22, 23).

In light of the fact that older adults may exhibit changes in nutritional status, sleep quality, and gustatory system, the aim of the current study is to determine the influence of sleep quality and nutritional status on gustatory function in elderly adults.’
